# Drivers and distribution of global ocean heat uptake over the last half century

Maurice F. Huguenin ®[1,2,3] ✉, Ryan M. Holmes[1,3,4,5] & Matthew H. England ®[1,2]

Since the 1970s, the ocean has absorbed almost all of the additional energy in the Earth system due to greenhouse warming. However, sparse observations limit our knowledge of where ocean heat uptake (OHU) has occurred and where this heat is stored today. Here, we equilibrate a reanalysis-forced ocean-sea ice model, using a spin-up that improves on earlier approaches, to investigate recent OHU trends basin-by-basin and associated separately with surface wind trends, thermodynamic properties (temperature, humidity and radiation) or both. Wind and thermodynamic changes each explain ~ 50% of global OHU, while Southern Ocean forcing trends can account for almost all of the global OHU. This OHU is enabled by cool sea surface temperatures and sensible heat gain when atmospheric thermodynamic properties are held fixed, while downward longwave radiation dominates when winds are fixed. These results address long-standing limitations in multidecadal ocean-sea ice model simulations to reconcile estimates of OHU, transport and storage.

The ocean plays a critical role in modulating the Earth's climate system and over the last 50 years it has taken up over 89% of the excess energy due to greenhouse warming[1–5]. Since the early 1990s, the rate of ocean warming has likely doubled[6]. However, our current understanding of the spatial distribution of ocean heat uptake (OHU) and storage is limited, not least because of sparse observations with large uncertainties, especially in sea-ice covered regions[7] and the deep ocean[3]. For example, reliable observations of ocean heat content (OHC) in the upper 2000 m only start in 2005 with the Argo program that covers 60°S–60°N[8]. Before 2005, good observations are only available in the upper 700 m from expendable bathythermographs[9] and from a few select deep ocean cruise ship measurements[10,11]. Observation-based studies therefore focus mainly on trends over much shorter time periods (e.g., since 2005[12] or since the early 1990s[13]).

Fully coupled atmosphere-ocean general circulation models and ocean-sea ice models simulate a complete representation of the global ocean and are now increasingly used to assess the OHC evolution. However, fully-coupled models from the Coupled and Flux-Anomaly-Forced Model Intercomparison Projects (CMIP[14] and FAFMIP[15] respectively) generally exhibit larger biases than ocean-sea ice models, and

simulate an internal climate variability that is independent of observations. Modelling studies have investigated recent trends mainly in idealised settings[15,16] or in coupled simulations with an independent climate variability[14,17]. In contrast, ocean-sea ice models are constrained by atmospheric fields from a reanalysis product, and therefore follow the observed trajectory of internal and forced climate variability.

Global climate models (both fully coupled and ocean-sea ice only) suffer from internal model drift due to errors in the representation of physical processes, and thus they require a spin-up to equilibrate their climate and minimise drift. In ocean-sea ice models, a common spin-up approach, used for the Ocean Model Intercomparison Project phase 2 (OMIP-2)[18], applies six repeat cycles of 1958–2018 atmospheric forcing from the Japanese reanalysis data set JRA55-do[19]. However, there are two limitations associated with this approach: (1) after each cycle, the model experiences a large shock and associated recovery period when the forcing suddenly switches from the year 2018 back to 1958 and (2) it is unclear how to account for model drift without a parallel running control simulation (Supplementary Fig. 1a).

In this study we address these limitations of the OMIP-2 approach by introducing a spin-up protocol for global ocean-sea ice models and

[1]Climate Change Research Centre, University of New South Wales, Sydney, NSW, Australia. [2]ARC Australian Centre for Excellence in Antarctic Science, University of New South Wales, Sydney, NSW, Australia. [3]ARC Centre of Excellence in Climate Extremes, University of New South Wales, Sydney, NSW, Australia. [4]School of Mathematics and Statistics, University of New South Wales, Sydney, NSW, Australia. [5]School of Geosciences, University of Sydney, Sydney, NSW, Australia. ✉e-mail: m.huguenin-virchaux@unsw.edu.au

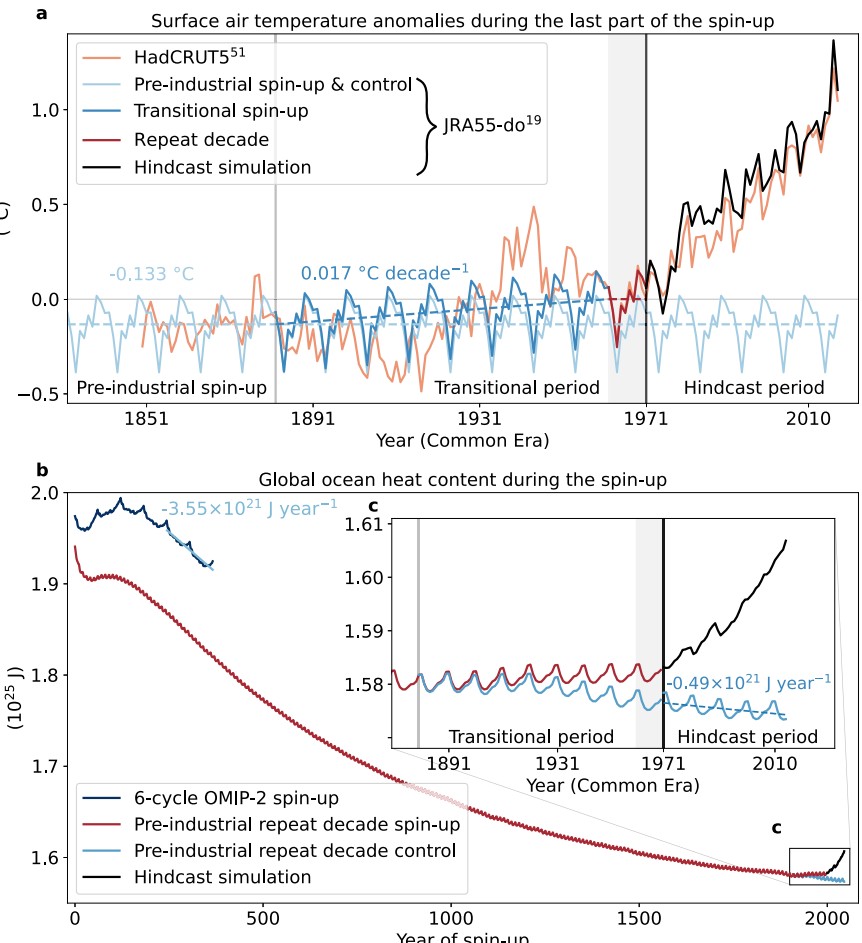

**Fig. 1 | Experimental design of the spin-up. a** Time series of global mean surface ocean air temperature anomalies from JRA55-do[19] during the last part of the ocean-sea ice model spin-up. The initial 1900 years of the spin-up are performed by applying repeat cycles of 1962–1971 atmospheric reanalysis forcing, from which a pre-industrial offset of 0.133 °C has been removed (light blue line and value). In orange the same anomalies from the observational data set HadCRUT5.1[51] which has a mean offset of 0.133 °C over 1850–1879 relative to the 1960s. During the transitional spin-up period, the offset increases by 0.017 °C decade⁻¹ (light blue value and linear trend) back to the 1960s level. This is to simulate the transition from the equilibrated pre-industrial to the warmer 1960s oceanic state. The 1962–1971 decade is shown as the grey shaded period. From 1972 onward, inter-annual hindcast simulations are then branched off (e.g., the full forcing simulation

in black where all atmospheric forcing fields evolve over time). The parallel control simulation is obtained by continuing the modified pre-industrial spin-up (light blue line) unchanged through the transitional period past 1972. **b** Time series of global ocean heat content during the five-cycle OMIP-2 spin-up (dark blue line, 10²⁵ J, and linear trend over the last two cycles, −3.55 × 10²¹ J year⁻¹) and the pre-industrial spin-up (red line, 10²⁵ J). The offset between the two time series at year 1 of the spin-up is due to the use of updated temperature fields and bathymetric changes in the repeat decade spin-up. **c** Inset of the last part of the spin-up, showing the transitional and hindcast periods with the 1960s period shaded in grey. The control simulation is given in light blue with its linear trend of −0.49 × 10²¹ J year⁻¹ over 1972–2017. The black line is the ocean heat content in the full forcing simulation initialised in 1971.

illustrate its benefits using the ACCESS-OM2 ocean-sea ice model[20]. The spin-up is performed using repeat decadal cycles of the JRA55-do reanalysis forcing from 1962–1971, corrected for pre-industrial times, to equilibrate the model to a state prior to the recent rapid acceleration in OHU (Fig. 1 and Methods). There are no longer large initial shocks at the beginning of each spin-up cycle and we can account for model drift by subtracting the linear trend from a parallel control repeat decade simulation (Fig. 1b, c). Using this approach in an observationally constrained model gives us an estimate of the actual trajectory of OHC, including the multi-decadal internal variability since the 1970s. By decomposing the atmospheric trends into processes and regions (Methods), we can attribute the global heat uptake by drivers and basins over this period.

## Results

### Global ocean heat uptake

The observations of upper 2000 m global OHC[3] reach 2.40 × 10²³ J in 2017 relative to the 1972–1981 baseline (dashed red line, Fig. 2a). We choose this baseline as it ends before the volcanic eruption of El

Chichón in mid-1982 and the OMIP-2 models prior to 1972 undergo a very strong global cooling period (Supplementary Fig. 2a). The multi-model mean from the fully coupled CMIP6 model suite (light blue line in Fig. 2a) tracks the observed OHC estimate closely, however with an increasingly large spread among ensemble members. The full forcing ACCESS-OM2 hindcast (where all atmospheric forcing evolves over time) simulates a global OHC increase of 1.73 × 10²³ J in the upper 2000 m (capturing 72% of the observational estimate).

This simulation improves considerably on the ACCESS-OM2 simulation that used the OMIP-2 spin-up approach, which lies at the bottom of the OMIP-2 ensemble (cf. black and dark blue lines in Fig. 2a). The hindcast also improves on most of the other 11 OMIP-2 models[18], whose multi-model-mean reaches 0.94 × 10²³ J in 2017, and captures a more realistic rise in OHC without the rapid spurious global cooling adjustment prior to 1972 (Supplementary Fig. 2a). There is no control simulation available to use for de-drifting in the OMIP-2 protocol, and we have attempted to de-drift the global OHC by fitting and removing a linear trend over the last two OMIP-2 cycles (e.g., black lines, Supplementary Fig. 1a). Without this de-drifting, the

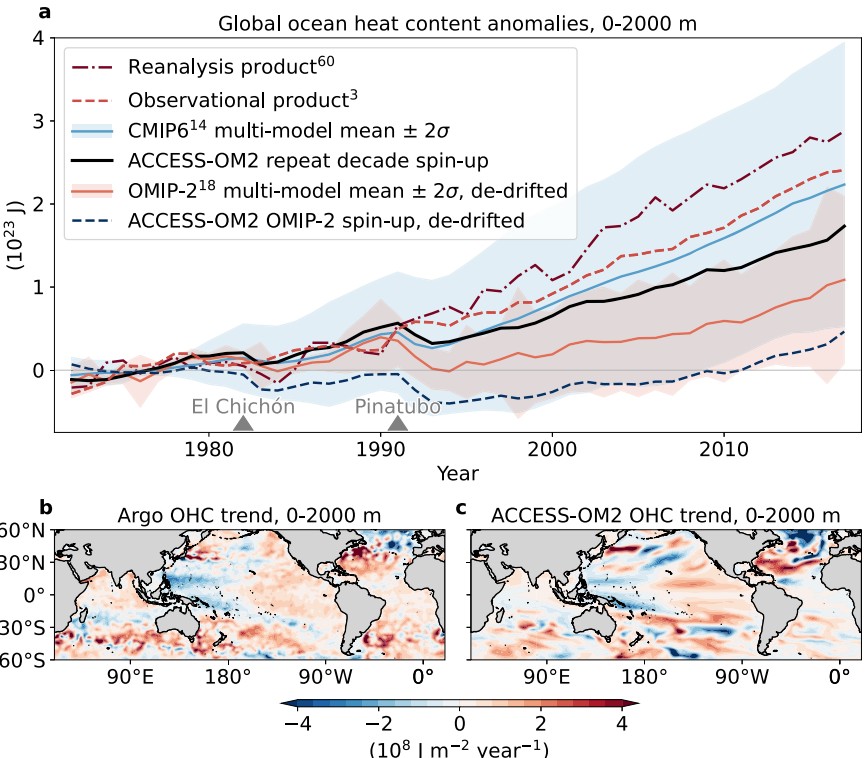

**Fig. 2 | Recent global ocean heat content (OHC) anomalies in observations and hindcast model simulations. a** Global ocean heat content anomalies (10²³ J) in the upper 2000 m from ocean reanalysis[60], observations[3], 25 fully coupled historical CMIP6 model runs[14] (including their multi-model mean and 2σ variance), the full forcing ocean-sea ice simulation (ACCESS-OM2 repeat decade spin-up, where all atmospheric forcing fields evolve over time), 11 de-drifted OMIP-2 ocean-sea ice model simulations[18] (including their multi-model mean, and 2σ variance) and the de-drifted ACCESS-OM2 OMIP-2-based simulation. For the individual time series of each CMIP6 and OMIP-2 ensemble member, see Supplementary Fig. 2. The two triangle markers highlight the volcanic eruptions of El Chichón in 1982 and Mount Pinatubo in 1991. The baseline period for all time series is 1972–1981. **b, c** Spatial distribution of anomalous upper 2000 m ocean heat content trends over 2006–2017 in the Argo observations and in the full forcing ACCESS-OM2 simulation (10⁸ J m⁻² year⁻¹).

positive trend in OHC in the OMIP-2 models would be even weaker (see also Fig. 24e in Tsujino et al.[18]). If a similar additive improvement, that we see in ACCESS-OM2, were applied to the other models in the OMIP-2 ensemble, then the multi-model mean of an ensemble using our alternative spin-up approach would reach an upper 2000 m OHC anomaly of $2.31 \times 10^{23}$ J in 2017, within four percentage points of the observations[3].

The spatial trend of the upper 2000 m OHC in the full forcing simulation corresponds well with Argo observations[21] (Fig. 2b, c and CMIP5 models over 2005–2015[22]), especially in the tropical Pacific and the Northern Atlantic. However, accumulation of anomalous heat in the model is reduced in the South Atlantic compared to Argo, and is likely caused by reduced ocean heat convergence in this region (see below). Most of the excess heat absorbed during this period is stored in the Southern Hemisphere (66.0% of the globally integrated trend relative to 72.7% in Argo). Over this shorter 2006-2017 period, the hemispheric OHC asymmetry has been linked to decadal climate variability[22], the asymmetry in anthropogenic forcing[23], the greater area of the Southern Hemisphere ocean[24] as well as anomalous ocean heat transport[12].

### Heat uptake, transport and storage rates

In order to quantify the spatial distribution of OHC trends, we consider the vertically integrated heat budget which expresses the OHC tendency (termed here heat storage) as the sum of the anomalous net surface heat flux (heat uptake) and the convergence of the anomalous vertically integrated ocean heat transport (Eq. (2), Methods). Globally integrated, the full-depth heat uptake/storage rate over the last half century in the full forcing simulation is $5.4 \times 10^{21}$ J year⁻¹ (Fig. 3a). While trends have accelerated over the last 20 years, the spatial pattern of

heat uptake has remained robust (cf., Fig. 3a and Supplementary Fig. 3a). The Southern Ocean dominates heat uptake with a rate of $6.9 \times 10^{21}$ J year⁻¹. The dominant role of this region is a consequence of the strong heat fluxes into the ocean where sea surface temperatures (SSTs) are colder than the overlying atmosphere. These cold SSTs are maintained by strong westerly winds that drive upwelling of cold water to the surface, insulating the Southern Ocean from forced changes, and driving efficient heat uptake from the atmosphere[17,25–27]. In this simulation, heat uptake occurs predominantly in the Indian and Pacific sectors of the Southern Ocean. Northward Ekman transport subsequently subducts these water masses along isopycnals into mode and intermediate water layers[27]. Heat storage is also significant in the Atlantic sector of the Southern Ocean where it arises primarily from the convergence of oceanic heat transport rather than from local atmospheric heat uptake (Fig. 3a, b).

Patterns of heat uptake outside of the Southern Ocean are more variable. Heat loss is dominant in the Atlantic basin ($-1.9 \times 10^{21}$ J year⁻¹), especially north of 45°N. The Atlantic heat loss arises from its connection to the Southern Ocean via the Atlantic Meridional Overturning Circulation (AMOC). The AMOC transports 42% ($2.9 \pm 0.2 \times 10^{21}$ J year⁻¹) of the additional heat taken up in the Southern Ocean northward into the Atlantic (red arrow in Fig. 3b), where two-thirds thereof is lost to the atmosphere via ocean-air heat fluxes. Compared to observations, the model's AMOC maximum at 26.5°N is weak (9.1 Sv relative to the observed estimate of 17 Sv over 2004–2012[28], 1 Sv = 10⁶ m³ s⁻¹, Supplementary Fig. 4a), lower than most other OMIP-2 models[18], and may thus lead to weaker anomalous Southern Ocean heat export into the Atlantic. However, the changes in the AMOC strength in the full forcing simulation of -1 Sv are small compared to the decadal variability of ± 2 Sv (black line, Supplementary Fig. 4a).

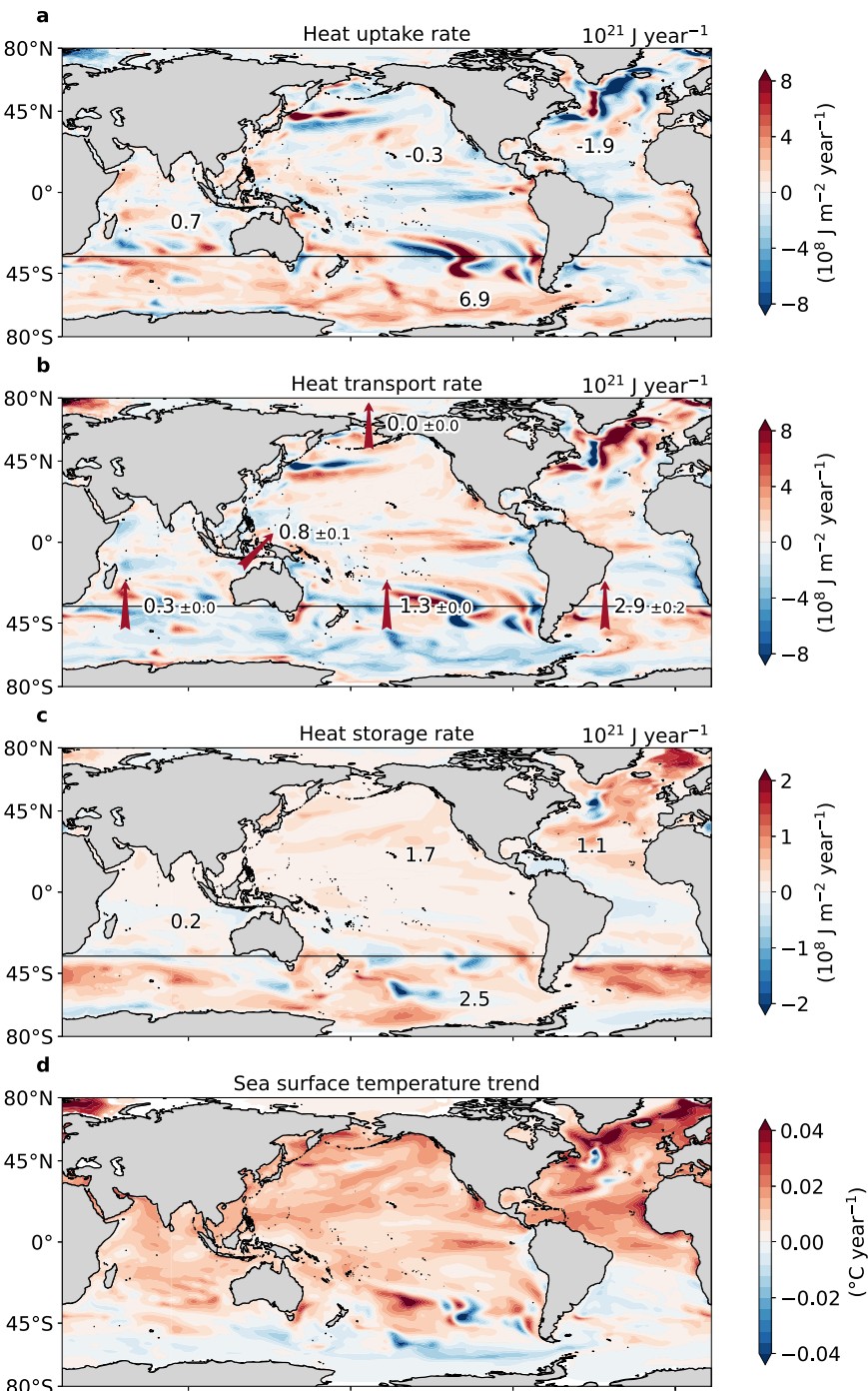

**Fig. 3 | Spatial distribution of ocean heat uptake, transport, storage and sea surface temperature trends over 1972–2017 in the full forcing simulation (where all atmospheric forcing fields evolve over time). a** Time integrated net surface heat flux anomalies ($10^8$ J m$^{-2}$ year$^{-1}$) with positive heat uptake defined as into the ocean. The basin-wide values ($10^{21}$ J year$^{-1}$) show the total area integrated trends over a particular ocean basin with the boundaries set by the black lines across the Southern Ocean, the Indonesian Throughflow, the Bering Strait and the continental land masses. The Southern Ocean ends at 36°S, the Bering Strait is at 65°N and the Indonesian Throughflow is defined between Java, New Guinea (105°W to 134°W) at 3°S and the Australian continent (20°S to 6°S) at 137°E. The Atlantic Ocean contributions include the Arctic Ocean north of 65°N and the marginal Hudson Bay, Baltic and Mediterranean basins. The Indian Ocean component also includes the Red Sea. The basin-wide values are rounded to one-decimal point accuracy. **b** Anomalous heat transport convergence calculated as a residual from the **a** heat uptake and **c** heat storage ($10^8$ J m$^{-2}$ year$^{-1}$). The anomalous heat transport rates and their uncertainties across transects ($10^{21}$ J m$^{-2}$ year$^{-1}$) are calculated from anomalous heat and volume transports (Methods). **d** Simulated sea surface temperature trends (°C year$^{-1}$). Grid cells in **d** that have a climatological sea ice coverage above 85% have been removed and are shaded white.

Heat uptake in the Indian and Pacific subtropical and tropical basins plays only a minor role on the global scale (Fig. 3a). This is likely because the Indian and Pacific basins lack a convection-driven deep circulation[29,30] that would efficiently take up heat over multi-decadal time scales. In addition, heat uptake in the tropics is inhibited by the warming response of the SST (Fig. 3d). In contrast, at the high latitudes of the Southern Ocean, the SST increases at a rate that keeps pace with local atmospheric warming (due to wind-driven

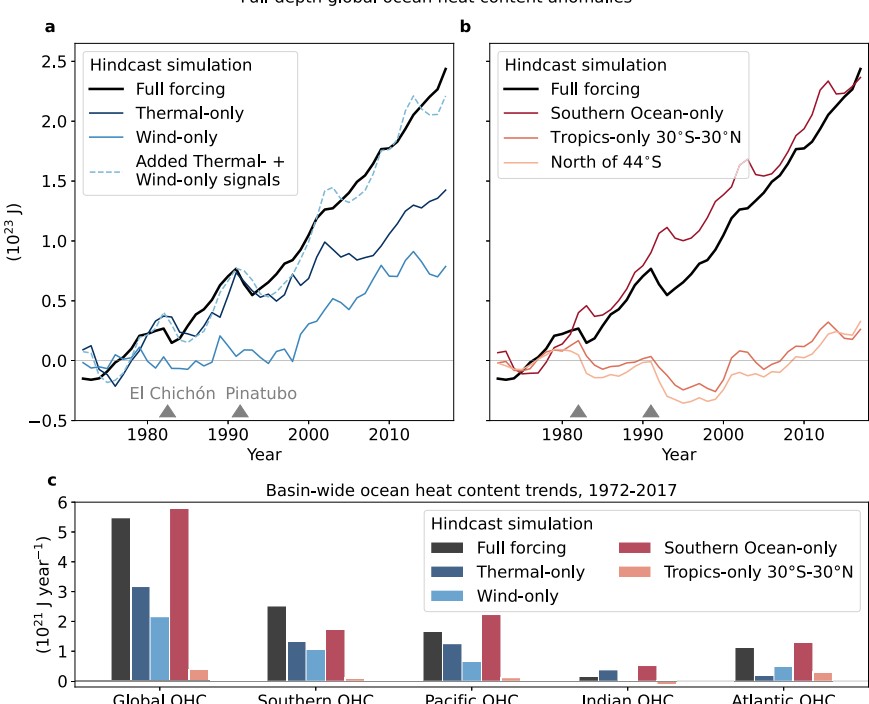

**Fig. 4 | Simulated global and regional ocean heat content (OHC) changes due to thermal/wind trends and due to regionally-constrained atmospheric trends. a** Time series of full-depth global ocean heat content anomalies ($10^{23}$ J) in the full forcing simulation (black line), when only prescribing surface wind trends (i.e., Wind-only) and when only prescribing thermodynamic trends (i.e., Thermal-only, Methods). The dashed blue line shows the anomalies in both wind- and thermal-only hindcast simulations added together. The two triangle markers highlight the volcanic eruptions of El Chichón in 1982 and Mount Pinatubo in 1991. The baseline period for all time series is 1972–1981. **b** Time series for the hindcast simulations where combined interannual wind and thermal forcing is applied only over the Southern Ocean (south of 44°S), the mid- and high northern latitudes (north of 44°S) and only over the tropics (30°S–30°N) with the remaining ocean area forced by the control repeat decade forcing. **c** Basin integrated ocean heat content trends ($10^{21}$ J year⁻¹) in the hindcast simulations of **a** and **b**.

Ekman effects) creating favourable conditions for continuous ocean heat uptake (Fig. 3d).

## Wind versus thermal effects

We next consider a set of hindcast simulations that isolate the impact of thermodynamic- (including air temperature, humidity and downward radiation) and wind-driven atmospheric changes over the global ocean and specific regions to better understand the drivers of recent OHU (Methods). In the wind-only simulation, zonal and meridional surface winds evolve over time while the other forcing fields are held fixed in the 1960s (and vice versa for the thermal experiment). The approach here differs from coupled and flux-anomaly forced ocean-sea ice model simulations that also aim to isolate contributions from winds and other changes[15,31] in that our experiments are forced by atmospheric trends from reanalysis instead of, for example, doubled atmospheric $CO_2$ concentrations, and thus they capture the observed trajectory of internal climate variability. The strong decadal variability in our simulations arises from the portion of the atmospheric forcing (whether thermal or wind forcing) that cycles through the repeat decade (Fig. 4a, b).

The two simulations that include only either thermal or surface wind trends explain 57% and 40% of the global OHC trend of $5.4 \times 10^{23}$ J (Fig. 4a, c). As in the full forcing simulation, heat uptake in both thermal- and wind-only experiments is dominated by the Southern Ocean (3.1 and $3.9 \times 10^{21}$ J year⁻¹, Supplementary Fig. 5a, e). In the wind-only simulation, Southern Ocean heat uptake is large because the SST cools as a result of enhanced northward Ekman transport of cool fresh Antarctic surface waters (Fig. 5a, b). This heat uptake is driven by sensible and upward longwave heat losses associated with the negative SST anomalies (Fig. 5c,d). Some compensation by latent and upward shortwave heat flux anomalies, due to increases in sea ice, are

associated with cooling in this region[32] (Supplementary Table 1). It is important to note that wind changes also have a direct impact on sensible and latent heat fluxes through their dependence on wind speed in the model's bulk formulae. As opposed to the wind-only experiment, heat uptake in the thermal-only experiment is associated mainly with changes in downward longwave radiation (Fig. 5c), which appear more important than air temperature changes (as the sensible heat flux anomalies are reduced). Integrated over the Southern Ocean, the sensible heat flux drives almost double the heat uptake than the longwave radiative flux in the wind-only simulation (3.7 vs. $1.9 \times 10^{21}$ J year⁻¹), while in the thermal-only simulation heat uptake through downward longwave radiation is more dominant (3.0 vs. $2.4 \times 10^{21}$ J year⁻¹, Supplementary Table 1).

Both changes in surface winds and atmospheric thermodynamic properties can affect the export of anomalous heat from the Southern Ocean into the Pacific, Indian and Atlantic basins via the meridional overturning circulation. In particular, in the wind-only simulation, anomalous heat export northward is stronger than in the thermal-only simulation, due to the stronger westerlies which in turn increase the Ekman transport and thus the Southern Ocean's overturning circulation (Supplementary Fig. 5b, f). In contrast, the parameterised submesoscale eddy mixing, eddy advection and diffusion schemes play a minor role in contributing to ocean heat transport changes into the Atlantic and Indo-Pacific. In a fully coupled framework, Liu et al.[33] showed that in response to quadrupled atmospheric $CO_2$ concentrations, the poleward-strengthened westerlies displace and intensify the Southern Ocean's meridional overturning circulation which results in anomalous heat transport divergence at 60°S and increased surface heat fluxes while the opposite was shown for 45°S. In our wind-only simulation, we see strong heat transport divergence at almost all latitudes of the Indian and Pacific sectors of the Southern Ocean, while

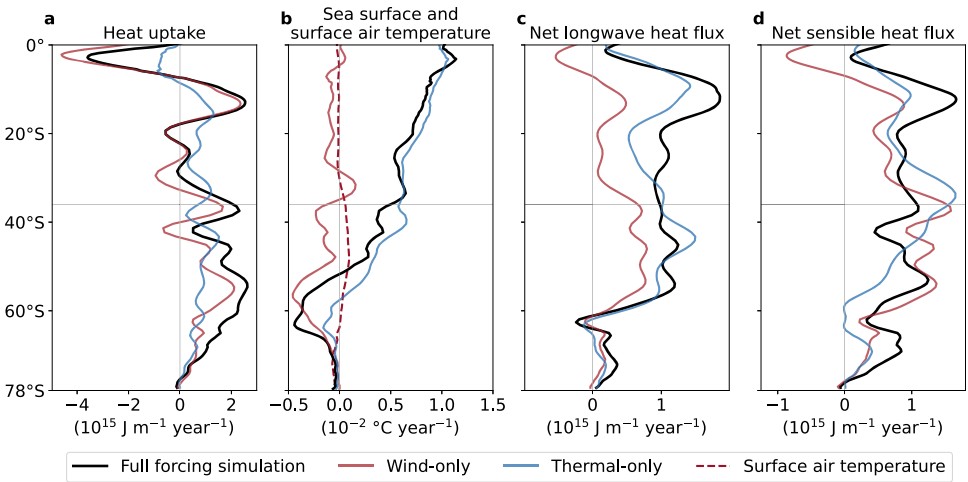

**Fig. 5 | Southern Hemisphere ocean heat uptake, sea surface temperature and surface air temperature, net longwave and net sensible heat flux trends over 1972–2017. a** Zonally integrated heat uptake in the simulations with full, wind-only and thermal-only forcing (10¹⁵ J year⁻¹), equal to the zonal integral of the spatial structure shown in Fig. 2a and Supplementary Fig. 5a, e. **b** Zonal mean sea surface temperature and surface air temperature trends (°C year⁻¹). **c, d** The contribution of net longwave and sensible heat fluxes to the total ocean heat uptake shown in **a** (10¹⁵ J year⁻¹). The horizontal lines at 36°S indicate the northern boundary of the Southern Ocean in our analysis. A 5-grid cell rolling mean has been applied in **a, c** and **d**.

heat converges in the Atlantic sector between 60°S-45°S (Supplementary Fig. 5b), likely because the Southern Ocean surface wind trends in JRA55-do are strongest in the Indian and Pacific sectors. We agree with Liu et al.[33], that wind stress changes are likely the primary drivers of ocean heat content change in the wind-only simulation (through their induced SST changes), rather than the direct wind-speed related turbulent heat flux change.

### Regional contributions
On the global scale, the OHC trend can be reproduced when atmospheric trends in both winds and thermodynamic properties are applied only over the Southern Ocean south of 44°S (with repeat decade forcing applied north of this latitude, Fig. 4b). However, an important regional difference between the full forcing and Southern Ocean-only forced simulation is that in the latter, heat storage is larger in the Pacific, Indian and Atlantic Oceans and smaller in the Southern Ocean (cf., black and dark red bars in Fig. 4c). This is likely caused by enhanced northward heat transport in the Southern Ocean-only experiment across 36°S, despite similar Southern Ocean heat uptake rates in both simulations (6.98 vs. 6.97 × 10²¹ J year⁻¹, Fig. 3a, b and Supplementary Fig. 6a, b). However, the heat transport rates in the Southern Ocean-only experiment are influenced by the tapering zones between the repeat decade and interannual forcing. In addition, the Pacific and Atlantic basins experience weak heat loss across the surface due to these basins being forced by the cooler 1960s atmosphere (Supplementary Fig. 6a).

Performing an experiment with interannual trends applied only north of 44°S or just over the tropics 30°S–30°N, shows a global OHC trend of 0.3–0.4 × 10²¹ J year⁻¹ (Fig. 4c). A positive trend, distinct from the repeat decade forcing oscillation, emerges only in the mid–1990s (light pink line, Fig. 4a), and is likely linked to the observed shift of the Interdecadal Pacific Oscillation into a negative phase. This favours La Niña-like conditions with increased trade winds and enhanced tropical heat uptake[34,35]. OHC trends over the 1992–2011 period from the tropical 30°S–30°N experiment appear mainly centred on the Equator in the western Pacific at 150 m depth (Supplementary Fig. 7), and are consistent with the observed trends over the same period[34]. A rapid increase in Indian Ocean heat content since the year 2000 has also been shown in observations[36] and occurs in a simulation with interannual trends restricted to only the Indian Ocean (not shown). This signal has been linked to the enhanced trade

winds that strengthened warm water transport across the Indonesian Throughflow since the early 2000s[36,37]. However, over the 50-year time period, Interdecadal Pacific Oscillation-related trade wind and OHC changes for the most part cancel each other out as this climate mode underwent a full oscillation[34,38]. Additional model experiments with the interannual atmospheric trend forcing only applied over individual ocean basins north of 44°S/35°S (Pacific-only/Indian- and Atlantic-only experiments, Methods) reveal only minor OHC trends (Supplementary Figs. 8, 9). This further emphasises the key role of the Southern Ocean in driving global ocean heat content trends over the past half century.

## Discussion
We have documented the evolution of ocean heat uptake, transport and storage over the last 50 years in a global ocean-sea ice model following a spin-up approach that improves on past simulations of OHC trends using the standard OMIP-2 protocol. The full forcing hindcast simulation considerably improves on the simulation with the same model but using the OMIP-2 spin-up, and reproduces the estimated trajectory of OHC in observations better than most OMIP-2 ensemble members. If the OMIP-2 project would follow the spin-up approach presented here, it is likely that both the multi-model mean and ensemble spread in Fig. 2a would shift upwards and better capture the observed trends.

Changes in surface winds and thermodynamic properties over the Southern Ocean each drive about half of the global heat uptake signal over the last half century (Fig. 6). These heat changes have important consequences for the zonal transport of the Antarctic Circumpolar Current with continued warming likely further accelerating the zonal flow[13]. As in the simulations with full or basin-wide forcing, heat uptake in the wind- and thermal-only experiments in the Indian and Pacific basins is minor, while the Atlantic Ocean is consistently losing heat across its surface (blue arrows, Fig. 6). In the full forcing as well as the wind- and thermal-only simulations, northward heat export from the Southern Ocean into the Atlantic dominates over export into the Indian and Pacific basins. While the Indo-Pacific plays only a minor role in multi-decadal heat uptake and storage, it can substantially impact global OHC trends over shorter periods through enhanced ocean heat uptake and reduced SST warming associated with the Interdecadal Pacific Oscillation[39] (e.g., during global warming hiatus periods such as from 2000–2009).

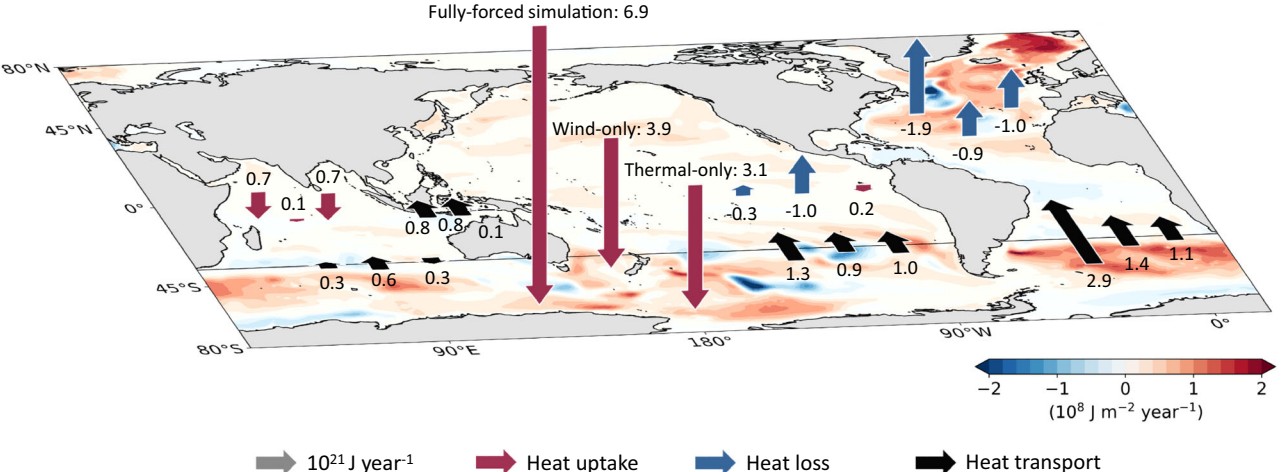

**Fig. 6 | Schematic summarising anomalous global ocean heat uptake, heat loss and heat transport over the last half century in different historical simulations.** The spatial pattern shows ocean heat storage rates in the full forcing simulation where all atmospheric forcing fields evolve over time ($10^8$ J m$^{-2}$ year$^{-1}$). The global ocean is divided into the Southern Ocean and the Indian, Pacific and Atlantic basins as in Fig. 3. The red and blue vertical arrows into and out of the plane show the basin integrated heat uptake and heat loss rates in the full forcing (left arrow), wind-only (middle arrow) and thermal-only (right arrow) simulations ($10^{21}$ J year$^{-1}$). The black arrows show the heat transport rates in the same simulations (from left to right: full, wind-only and thermal-only forcing) across the transects that separate the basins ($10^{21}$ J year$^{-1}$). The arrows are to scale, and values are rounded to one-decimal point accuracy. The transport rates across the Bering Strait are one magnitude smaller and not shown.

Over the last twenty years of the full forcing simulation, the weakening AMOC in the North Atlantic (Supplementary Fig. 4) may be linked to positive redistribution feedbacks that have been previously described in a coupled climate model[40]. In this feedback, a weakened AMOC decreases meridional heat transport in the North Atlantic, leading to a divergence of heat, cooler SSTs and increased heat uptake in the subpolar gyre, which in turn further weakens the AMOC[40,41]. It is unclear if this feedback mechanism is contributing to the North Atlantic changes in the full forcing simulation, as heat uptake north of the Equator decreases ($-0.6 \times 10^{21}$ J year$^{-1}$) and heat transport increases ($+0.6 \times 10^{21}$ J year$^{-1}$) over the last twenty years of the run, compared to the full period.

Limitations in our results arise from the use of a single model with a 1° horizontal resolution, the biases related to errors in the model's representation of physical processes and uncertainties in reconstructing past atmospheric forcing. Uncertainties also arise from inherent uncertainties in the reanalysis product used, including the reliability of the implied radiative heat flux trends due to both greenhouse gases and aerosols, which remain poorly constrained in observations. Heat transport and heat loss across the surface can be dependent on the model resolution[42] with biases expected to decrease in a finer grid[18]. However, the model configuration used here matches the typical resolution of most OMIP-2 and CMIP6 ensemble members, and heat content anomalies following the OMIP-2 protocol are similar when using the higher resolution configurations of the model (Supplementary Fig. 1b). The low computational cost of the model we employ also allowed us to minimise deep ocean model drift with a long spin-up and permitted a suite of multi-decadal simulations that would otherwise be too expensive to explore using higher-resolution models.

In summary, our experiments emphasise that recent trends in Southern Ocean surface winds, surface air temperature and radiation have driven almost all of the globally integrated ocean warming of the past half century. Increased observational coverage over the Southern Ocean is therefore key to reconcile global surface heat fluxes, ocean heat uptake and heat content changes, as well as building increased confidence in climate models and climate change projections for the coming decades.

## Methods
### Model, forcing and spin-up
We use the global ocean-sea ice model ACCESS-OM2[20] in a 1° horizontal resolution configuration with 50 z* vertical levels. ACCESS-OM2 consists of the Geophysical Fluid Dynamics Laboratory MOM5.1 ocean model[43] coupled to the Los Alamos CICE5.1.2 sea ice model[44] via OASIS3-MCT[45]. Atmospheric forcing for the model is derived from a prescribed atmospheric state using the Japanese Reanalysis product JRA55-do-1-3[19] which covers the period 1958–2018. The forcing fields are zonal and meridional wind speed, air temperature and specific humidity at 10 m as well as downward short- and longwave radiation, rain- and snowfall, river and ice-related runoff and sea level pressure at the ocean's surface. These fields are used to calculate zonal and meridional wind stress, surface heat and freshwater fluxes using bulk formulae[46]. More details on the model setup and performance can be found in Kiss et al.[20].

We perform a 2000-year spin up of the model initiated from World Ocean Atlas 2013 v2 conditions[47] using modified repeat cycles of the JRA55-do 1962–1971 decade. We choose this decade as it has no extreme El Niño-Southern Oscillation events or tendencies[48] and has close to neutral conditions in the Interdecadal Pacific Oscillation (IPO index: −0.1)[49]. However, it has a positive Southern Annular Mode and three positive Indian Ocean Dipole events occurred in this period[50]. The choice of this decade is a compromise between an early period with limited observations where our confidence in the atmospheric forcing is low, and later periods where the anthropogenic signal is larger and the hindcast experiments would be shorter.

For the first 1910 years of the spin-up, we subtract from the repeat 1962–1971 forcing a pre-industrial offset of 0.133 °C from the surface air temperature and 0.7 W m$^{-2}$ from the downward longwave radiation fields. This is to equilibrate the model to an estimate of the pre-industrial climate instead of a 1960s climate that already incorporates an anthropogenic footprint. Additionally, we modify the specific humidity in order to keep the relative humidity constant and avoid overly impacting evaporation and the latent heat flux. The surface air temperature offset is calculated from the difference between the JRA55-do mean during the 1962–1971 period and the years 1850–1879 in the HadCRUT5[51] data set (light blue and orange lines, Fig. 1a). The

offset in downward longwave radiation is consistent with values presented in the fifth Assessment Report of the Intergovernmental Panel for Climate Change (IPCC AR5, Fig. SPM.5)[4]. The overall ratio of surface air temperature to downward longwave radiation offsets is the same as in the study by Stewart and Hogg[52] where they used offsets derived from the CMIP5 historical and moderate greenhouse gas emission scenario (RCP4.5) to run idealised climate change hindcast experiments. As in IPCC AR5 Fig. SPM.5[4], the uncertainty in the pre-industrial offset of downward longwave radiation (and surface air temperature) is likely as large as the value itself, but it is a reasonable approach given the limited data available from pre-industrial times.

The period 1910–2000 of the spin-up (i.e., 1882–1971 Current Era) is the transitional period where we linearly reduce the offsets in the forcing fields back to 1962–1971 levels (dark blue and dark red lines, Fig. 1a). This represents the developing anthropogenic impact on the ocean between the pre-industrial state and the warmer 1960s climate. In year 2000 of the spin-up (i.e., year 1972 Current Era), the interannually-forced hindcast simulations begin. The control simulation is a continuation of the pre-industrial spin-up with modified repeat decade forcing beyond 1972 (light blue line, Fig. 1a).

### Hindcast experiments

We run a set of simulations that combine both climatological (1962–1971) and interannual (1971–2017) forcing to investigate the contribution of changing surface winds, thermodynamic properties and the role of individual oceanic regions to anomalous heat uptake since the 1970s.

The wind and thermal simulations include forcing the model over 1972–2017 with interannual zonal and meridional surface wind trends (the wind-only experiment) or combined surface air temperature, humidity, radiation, freshwater and sea level pressure trends (the thermal-only experiment), while repeat decade forcing is used for the other forcing fields. The hindcast experiments here do not allow a complete separation between buoyancy effects (including heating) and wind effects because buoyancy and heat fluxes both change in each of the wind- and thermal-only experiments; for example, the winds can force an SST change that will feed back and alter the sensible heat flux fields. Likewise the thermally-forced experiment can include changes in wind stress wherever ocean circulation changes are simulated, because the wind stress is controlled by the difference between wind speed and ocean current speed, although this effect is generally second order. While surface air temperature and radiation variations dominate the signal in the thermal-only simulation, freshwater fluxes can also contribute to changes in ocean circulation and thus ocean heat uptake and redistribution via changes in, for example, the meridional overturning circulation in the Atlantic and Southern Oceans[15,53].

The regional simulations (hereafter Southern Ocean-only, North of 44°S, Tropics-only 30°S–30°S, Pacific-, Indian- and Atlantic-only forcing simulations) include applying interannual trending atmospheric fields over a specific region of the global ocean while repeat decade forcing is applied over the remaining ocean area (e.g., blue contours in Supplementary Fig. 9). For these simulations, a linear smoothing boundary region of 4° latitude/longitude is used to combine the two forcing fields. For the Southern- and Pacific Ocean-only simulations, we choose the boundaries at 44°S as this latitude marks the poleward extent of the shallow subtropical cells. For the Indian and Atlantic Ocean simulations, we set the southern interannual forcing/repeat decade forcing boundary to 35°S at the southern tip of Africa.

### Ocean heat content calculations

Heat content,

$$H = \iiint \rho_0 C_p \Theta \, dV, \tag{1}$$

is calculated using a reference density $\rho_0 = 1035$ kg m$^{-3}$, a specific heat capacity $C_p = 3992.1$ J kg$^{-1}$ K$^{-1}$, the model's prognostic temperature variable Conservative Temperature $\Theta$[54,55] (K) and the (time-variable) grid cell volume d$V$ (m$^3$).

The vertically integrated Eulerian heat budget can be expressed as

$$\frac{\partial}{\partial t} \int_z^0 H \, dt = Q_{net} - \nabla_h \cdot \mathbf{F}, \tag{2}$$

where the left-hand side is the depth integrated heat content tendency at a given location (J m$^{-2}$ year$^{-1}$) between depth $z$ and the surface, $Q_{net}$ is the net surface heat flux and $\nabla_h \cdot \mathbf{F}$ is the divergence of the vertically integrated ocean heat transport. Changes in heat content arise from changes in heat exchange with the atmosphere (heat uptake) and/or from changes in the convergence of horizontal ocean heat transport. The anomalous heat uptake rate is calculated by first time integrating the net surface heat flux tendencies, including the turbulent (latent and sensible), radiative (short- and longwave), surface volume flux-associated and sea ice exchange components, before removing the linear trend in the time integrated tendencies of the control simulation, and finally fitting a linear trend to the result. The heat storage rate is calculated similarly. The heat transport convergences are calculated as the residual between heat uptake and storage (Eq. (2)). These calculations would be more difficult without a parallel-running control simulation (not available as part of OMIP-2) that can be used to remove drift as well as the steady-state pattern of heat input at low-latitudes and heat loss at high-latitudes connected by meridional ocean heat transport.

Ocean heat transport (OHT) rates across individual transects are calculated from the vertical integral of horizontal advective and parameterised diffusive, mesoscale- and submesoscale heat fluxes accumulated online. Uncertainties in these heat transport rates arise from the presence of non-zero net volume fluxes, which result in a dependence of the cross-transect heat transport on the arbitrary reference temperature[56,57]. We estimate the uncertainty in the anomalous heat transport rate based on the change in the volume transport across the transect $\Delta\Psi$ (m$^3$ s$^{-1}$) and a maximum possible range for the temperature $(\Delta\Theta)^{max}$ at which that net volume transport could be assumed to return:

$$\Delta OHT = \pm \rho_0 C_p \frac{(\Delta\Theta)^{max}}{2} \Delta\Psi. \tag{3}$$

We define $(\Delta\Theta)^{max}$ to be 30°C, an estimate of the maximum temperature range of the model. For example, if the maximum temperature of water transported through the Indonesian Throughflow is 30 °C, then the maximum ambiguity in the change in heat transport is estimated by assuming that this water returns back into the Pacific via the Southern Ocean at 0°C. This issue is discussed in more detail in Section S3 in the Supporting Information of Holmes et al.[56] and in Forget and Ferreira[57].

### CMIP6 products

To compare the simulations in this study to atmosphere-ocean general circulation models, we analyse 16 ensemble members from CMIP6 as shown in the Supplementary Table 2. The choice of the models and anomaly calculation is based on Irving et al.[58] and includes first taking a cubic fit of the globally integrated 0–2000 m OHC over the length of the pre-industrial control simulation in each model. The length of this control simulation can be between 500–6000 years depending on the model. This fit is then subtracted from the historical simulation (ending in 2014) and SSP5-8.5 (2014–2017) projection simulation before the removal of the baseline 1972–1981 period.

## Data availability

The model data to recreate the figures in this study have been deposited online in the Zenodo database under https://doi.org/10.5281/zenodo.6873094[59]. The full model output is stored on the National Computational Infrastructure and available upon contact to the first author. The Argo data were collected and made freely available by the International Argo Program and the national programs that contribute to it (http://www.argo.ucsd.edu, http://argo.jcommops.org). The Argo Program is part of the Global Ocean Observing System (https://doi.org/10.17882/42182). The product we used here was produced at the China Argo Real-time Data Center and available at http://www.argo.org.cn/english/. The CMIP6 data is available at the Earth System Grid Federation: https://esgf-node.llnl.gov/projects/cmip6/.

## Code availability

The analysis scripts to create the forcing for the JRA55-do-1-3 repeat decade spin-up and to reproduce the figures are published online in the Zenodo database under https://doi.org/10.5281/zenodo.6873094[59].

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

## Acknowledgements

We thank Hiroyuki Tsujino and Hakase Hayashida for providing access to the OMIP-2 model output, Damien Irving and Taimoor Sohail for calculating the CMIP6 time series and we thank the CMIP and Argo projects for providing access to their data sets. We thank the Consortium for Ocean-Sea Ice Modelling in Australia (COSIMA; www.cosima.org.au) for their technical help, continued development and for making the ACCESS-OM2 model available at https://github.com/COSIMA/access-om2. The simulations in this project were conducted with resources and services from the National Computational Infrastructure which is supported by the Australian Government. M.F.H. is supported by a Scientia PhD scholarship from the University of New South Wales (Program code 1476), R.M.H. is supported by the Australian Research Council (ARC) grant DE21010004 and M.H.E. is supported by the ARC Australian Centre for Excellence in Antarctic Science and is also supported by the Centre for Southern Hemisphere Oceans Research (CSHOR), a joint research centre between QNLM, CSIRO, UNSW and UTAS. Additionally, this work was supported by both the ARC Australian Centre for Excellence in Antarctic Science (ACEAS; ARC Grant No. SR200100008) and the ARC Centre of Excellence for Climate Extremes (CLEX; ARC Grant No. CE170100023).

## Author contributions

M.F.H. performed the analyses and wrote the intial draft of the paper in discussion with R.M.H. and M.H.E. All authors formulated the experimental design, contributed to interpreting the results and refinement of the paper.

## Competing interests

The authors declare no competing interests.

## Additional information

**Supplementary information** The online version contains

supplementary material available at https://doi.org/10.1038/s41467-022-32540-5.

