## [Peer Review File · Nature Communications]

Drivers and distribution of global ocean heat uptake over the last half centuryREVIEWER COMMENTS

Reviewer #1 (Remarks to the Author):

The authors performed ACCESS-OM2 hindcast simulations, which are initialized from an equilibrated spin-up run and forced by atmospheric fields constrained by observations. The hindcast simulations well simulate the observed ocean heat uptake since 1970s over global oceans wherein the Southern Ocean takes most of the heat. Their “wind” and “thermal” experiments further reveal the contributions of either forcer to global and regional ocean heat uptake and associated physical mechanisms.

This is a nice study that contributes to the understanding of global ocean heat uptake and redistribution during the past several decades. It features a novel equilibrated spin-up, which allows the authors’ hindcast simulations to better simulate global ocean heat content changes than most previous OMIP-2 simulations. I also like the accompanying “wind” and “thermal” experiments and associated comprehensive analysis. Thereby, I would like to recommend publication of this manuscript pending on minor revisions.

1. I am wondering whether it would be better to name the “buoyant” experiment for the “thermal” experiment. This is because, as per “Method”, the “thermal” experiment is forced by combined surface air temperature, humidity, radiation, precipitation and sea level pressure trends in which freshwater forcing (e.g., precipitation) is also included. Moreover, the change in sea ice may also induce freshwater flux into the ocean. All these freshwater flux changes may contribute to the alteration of ocean circulation, especially the meridional overturning circulations (MOCs) in the Atlantic and Southern Oceans, which could significantly influence ocean heat uptake and redistribution (e.g., Winton et al. 2013; Gregory et al. 2016). Just for the authors’ reference.

2. The authors may want to briefly discuss the “North Atlantic redistribution feedback” (Couldrey et al. 2021; Liu et al. 2020) that plays an important role in the Atlantic heat uptake and distribution. In this feedback, a weakened AMOC diminishes the northward heat transport and generate a meridional divergence of oceanic heat transport in the subpolar North Atlantic, which leads to a cooler SST there and enhanced ocean heat uptake. The enhanced ocean heat uptake further contributes to the weakening of the AMOC. This feedback has been demonstrated in a fully coupled framework (Liu et al. 2020) wherein the enhanced ocean heat uptake act to dump the high-latitude cooler SST primarily through changes in turbulent heat fluxes (via ocean-atmosphere interaction). I understand the current study is primarily based on an ocean-sea ice model while a brief discussion of above feedback might be desirable.

3. The authors may want to briefly discuss the change in Southern Ocean MOCs which potentially affect Southern Ocean heat uptake and distribution. The wind-driven MOC is tied to Ekman transport but this circulation extends toward the full-depth ocean. Thereby, it might be more straightforward to link the MOC change to the change in ocean heat transport in the vertically integrated Eulerian heat budget. Meanwhile, mesoscale- and sub mesoscale ocean eddies, diffusion and mixing processes also contribute to the changes in MOC and/or ocean heat transport at different latitudes in the Southern Ocean, which could be further linked to the ocean heat transport rates across individual transects as illustrated in the paper. On the other hand, the “wind” and “thermal” experiments might help shed lights on the roles of both forcers in above processes. For example, in a fully coupled framework, Liu et al (2018) showed that the poleward-strengthened zonal surface winds in response to atmospheric CO₂ increase displace and intensify the Deacon cell and thus the residual MOC in the Southern Ocean, resulting to an anomalous divergence of meridional oceanic heat transport around 60°S coupled to a surface heat flux increase but an anomalous convergence of oceanic heat transport and heat loss at the surface. They also showed that the wind effect are primarily through altering ocean circulation while the surface wind speed change, as involved in the bulk formula of turbulent heat fluxes, has a minimal influence on the ocean heat uptake and redistribution over the Southern Ocean.

References not cited in the manuscript

1. Couldrey, M.P., Gregory, J.M., Boeira Dias, F., Dobrohotoff, P., Domingues, C.M., Garuba, O., Griffies, S.M., Haak, H., Hu, A., Ishii, M. and Jungclaus, J., 2021. What causes the spread of model projections of ocean dynamic sea-level change in response to greenhouse gas forcing?. *Climate Dynamics*, 56, 155-187.
2. Liu, W., Fedorov, A.V., Xie, S.P. and Hu, S., 2020. Climate impacts of a weakened Atlantic Meridional Overturning Circulation in a warming climate. *Science Advances*, 6, eaaz4876.
3. Liu, W., Lu, J., Xie, S.P. and Fedorov, A., 2018. Southern Ocean heat uptake, redistribution, and storage in a warming climate: The role of meridional overturning circulation. *Journal of Climate*, 31, 4727-4743.
4. Winton, M., Griffies, S.M., Samuels, B.L., Sarmiento, J.L. and Frölicher, T.L., 2013. Connecting changing ocean circulation with changing climate. *Journal of climate*, 26, 2268-2278.

Reviewer #2 (Remarks to the Author):

This paper presents a 1972-2017 hindcast of ocean heat uptake and details where ocean heat content has been stored and transported. It goes without saying improved estimates of these metrics are of great scientific importance and worthy of publication in *Nature Communications*. This especially is the case for the pre-argo era where observations are sparse. However there presently exists a large number of published observations, re-analysis and modeling-based estimates of these quantities (as the authors reference in the paper). Previous work also reaches the same key conclusion that the Southern Ocean dominates heat uptake and storage (e.g., Frölicher et al. 2015). So, one important question is why should we place emphasis on the results and conclusions presented by the authors here over the previous modeling work? Here the paper stumbles. It certainly emphasizes what is poor about previous efforts, but in the main text itself presents very little convincing arguments about what is superior about using re-analysis to drive the model and the decadal spin up technique. For that matter, in just reading the main text one would have little idea about what the new technique even entails (see my comment on line 45). Therefore, I would argue that presenting a strong case for the superiority of their hindcast compared to previous work is the lynchpin of the study, as how much we care about the excellently presented results that follows depends upon the novelty and superiority of these techniques used to create them. In the methods sections and the SI a case is built, but I think that this information has to be presented more as a result in the paper not just in the methods section.

Another strength of this paper lies in the subset of both mechanistic (thermal vs wind) and regional hindcast runs. In particular, the Southern Ocean only forced experiment basically reproducing the global heat storage of the global forced experiment strikes me as a very important result. However, the fact that the study contains the regional experiments barely gets mentioned in the abstract. So perhaps more emphasis on the experiments performed in both the abstract and introduction will also differentiate this work more from past modelling studies.

Both of these comments are largely stylistic, and I feel that the material submitted (i.e., main text+SI) can largely be altered to address them without the need for additional experiments to be performed. Aside from the caveats associated with the fact that the results are produced by an ocean model driven by a reanalysis product, the scientific analysis and presentation seem accurate and clear. To that end I would recommend acceptance after minor revisions.

Comments:

Line 41: If it is independent, why do you care about removing drift later on.

Line 45: Evidence is needed in the main text to explain and show that the new spin-up technique of using repeat decades is better than previous efforts. After all, the quality of the results presented in this paper depend upon this.

Line 56: Reaches 2.40E23J by which year? The term 'observational record' is rather vague. Which observations exactly?

Line 62: Please define in the main text what is meant by full forcing. At this stage of reading, I have no idea what re-analysis dataset is being used or what is even meant by forcing (perhaps though this is more clearly stated at line 119). I have no problem about things being in the method section, but the novelty of the paper and results themselves lie in these techniques. A more solid description is needed in the main text (to repeat my comment above!)

Line 66: Could not the improvement be coincidental. Why does it improve?

Line 77: How do the CMIP5 models go to 2017 or for that matter CMIP6 which stops in 2014?

Figure 1: Levitus 2020 (line 75) or 2012 (figure key)?

Figure 2: Just to confirm I see just one blue line here demarking the Southern Ocean (not lines as the caption says). Should the continental land masses be used to separate the other ocean basins?

Line 125: I agree with the sentiment here, but you are still subject to the quality of the reanalysis surface fluxes and (for example) if they properly capture radiative forcing trends due to greenhouse gases and aerosol. Have you thought about what might happen if you said used ERA5 or another product?

Line 178: I think the conclusion here supports my claim that you should devote at least some of the main text (not methods) on explaining and evaluating the spin-up technique.

Line 243: This is an example of some text, which details the improvement, that would be better off in the main text.

Response to Reviewers

Ref.: NCOMMS-22-03243

All line and figure numbers in blue point to the manuscript with tracked changes active.

Reviewer #1:

The authors performed ACCESS-OM2 hindcast simulations, which are initialized from an equilibrated spin-up run and forced by atmospheric fields constrained by observations. The hindcast simulations well simulate the observed ocean heat uptake since the 1970s over global oceans wherein the Southern Ocean takes most of the heat. Their “wind” and “thermal” experiments further reveal the contributions of either forcer to global and regional ocean heat uptake and associated physical mechanisms.

This is a nice study that contributes to the understanding of global ocean heat uptake and redistribution during the past several decades. It features a novel equilibrated spin-up, which allows the authors’ hindcast simulations to better simulate global ocean heat content changes than most previous OMIP-2 simulations. I also like the accompanying “wind” and “thermal” experiments and associated comprehensive analysis. Thereby, I would like to recommend publication of this manuscript pending on minor revisions.

We thank the reviewer for their useful and informative comments that have helped to improve the manuscript. Below we address each comment individually.

1. I am wondering whether it would be better to name the “buoyant” experiment for the “thermal” experiment. This is because, as per “Method”, the “thermal” experiment is forced by combined surface air temperature, humidity, radiation, precipitation and sea level pressure trends in which freshwater forcing (e.g., precipitation) is also included. Moreover, the change in sea ice may also induce freshwater flux into the ocean. All these freshwater flux changes may contribute to the alteration of ocean circulation, especially the meridional overturning circulations (MOCs) in the Atlantic and Southern Oceans, which could significantly influence ocean heat uptake and redistribution (e.g., Winton et al. 2013; Gregory et al. 2016). Just for the authors’ reference.

Yes we thought long and hard about the best experiment names prior to submission. In the end we decided to define our simulations as the full forcing, wind-only and thermal-only simulation. This is because our experimental separation between “thermal-” and “wind-only” experiments here is distinct from a full separation between net air-sea buoyancy fluxes and wind stress effects. For example, the net air-sea buoyancy (and heat) flux fields change in both the wind and thermal experiments. It’s just that the primary change applied is either thermal or wind-driven in nature. Nonetheless to be clearer on this distinction, we have clarified on line 322 of the Methods section:

“The hindcast experiments here do not allow a complete separation between buoyancy effects (including heating) and wind effects because buoyancy and heat fluxes both change in each of the wind- and thermal-only experiments; for example, the winds can force an SST change that will feed back and alter the sensible heat flux fields. Likewise the thermally-forced experiment can include changes in wind stress

wherever ocean circulation changes are simulated, because the wind stress is controlled by the difference
between wind speed and ocean current speed, although this effect is generally second order.”

We also added the additional information on the impact of freshwater fluxes in the Methods section when
describing the experimental design of the perturbation simulations (lines 330ff.):

“While surface air temperature and radiation variations dominate the signal in the thermal-only
simulation, freshwater fluxes can also contribute to changes in ocean circulation and thus ocean heat uptake
and redistribution via changes in, for example, the meridional overturning circulation in the Atlantic and
Southern Oceans (Winton et al., 2013; Gregory et al., 2016).”

When settling on these experiment names, we also felt the term *thermal* is easier to understand for the broader
*Nature Communications* community, and remains apt given the dominance of thermal effects, while the term
*buoyancy* is more specifically used in the physical oceanography community. In any case, we hope the revised
text provides suitable clarity on this naming convention.

2. The authors may want to briefly discuss the “North Atlantic redistribution feedback” (Couldrey et al. 2021;
Liu et al. 2020) that plays an important role in the Atlantic heat uptake and distribution. In this feedback, a
weakened AMOC diminishes the northward heat transport and generates a meridional divergence of oceanic
heat transport in the subpolar North Atlantic, which leads to a cooler SST there and enhanced ocean heat
uptake. The enhanced ocean heat uptake further contributes to the weakening of the AMOC. This feedback
has been demonstrated in a fully coupled framework (Liu et al. 2020) wherein the enhanced ocean heat
uptake act to dump the high-latitude cooler SST primarily through changes in turbulent heat fluxes (via ocean-
atmosphere interaction). I understand the current study is primarily based on an ocean-sea ice model while a
brief discussion of above feedback might be desirable.

Thank you for highlighting this positive feedback mechanism of a weakening AMOC and heat
uptake/divergence in the subpolar gyre of the North Atlantic. We have now included a short description of this
feedback mechanism in the discussion section in lines 247ff:

“Over the last twenty years of the full forcing simulation, the weakening AMOC in the North Atlantic
(Extended Data Fig. 4) may be linked to positive redistribution feedbacks that have been previously described
in a coupled climate model (Liu et al., 2020). In this feedback, a weakened AMOC decreases meridional heat
transport in the North Atlantic, leading to a divergence of heat, cooler SSTs and increased heat uptake in the
subpolar gyre, which in turn further weakens the AMOC (Couldrey et al., 2021; Liu et al., 2020). This feedback
mechanism is likely contributing to the North Atlantic changes in the full forcing simulation, with heat uptake
increasing ($+0.6 \times 10^{21}$ J year⁻¹) and heat transport decreasing (-0.6×10^{21} J year⁻¹) over the last twenty years
of the run, compared to the last half century.”

3. The authors may want to briefly discuss the change in Southern Ocean MOCs which potentially affect
Southern Ocean heat uptake and distribution. The wind-driven MOC is tied to Ekman transport but this
circulation extends toward the full-depth ocean. Thereby, it might be more straightforward to link the MOC
change to the change in ocean heat transport in the vertically integrated Eulerian heat budget. Meanwhile,
mesoscale- and sub mesoscale ocean eddies, diffusion and mixing processes also contribute to the changes
in MOC and/or ocean heat transport at different latitudes in the Southern Ocean, which could be further linked

to the ocean heat transport rates across individual transects as illustrated in the paper. On the other hand,
 the “wind” and “thermal” experiments might help shed light on the roles of both forcings in above processes.
 For example, in a fully coupled framework, Liu et al (2018) showed that the poleward-strengthened zonal
 surface winds in response to atmospheric CO₂ increase displace and intensify the Deacon cell and thus the
 residual MOC in the Southern Ocean, resulting to an anomalous divergence of meridional oceanic heat
 transport around 60°S coupled to a surface heat flux increase but an anomalous convergence of oceanic heat
 transport and heat loss at the surface. They also showed that the wind effects are primarily through altering
 ocean circulation while the surface wind speed change, as involved in the bulk formula of turbulent heat fluxes,
 has a minimal influence on the ocean heat uptake and redistribution over the Southern Ocean. ^[1]_{SEP}

Thank you for this detailed explanation. We have now included this information on changes of the Southern
 Ocean overturning circulation in the section where we present the wind-only and buoyancy-only simulations
 (lines 180ff.):

“Both changes in surface winds and atmospheric thermodynamic properties can affect the export of
 anomalous heat from the Southern Ocean into the Pacific, Indian and Atlantic basins via the meridional
 overturning circulation. In particular, in the wind-only simulation, anomalous heat export northward is stronger
 than in the buoyancy-only simulation, due to the stronger westerlies which in turn increase the Ekman
 transport and thus the Southern Ocean's overturning circulation (Fig. S4b, f). In contrast, the parameterised
 submesoscale eddy mixing, eddy advection and diffusion schemes play a minor role in contributing to ocean
 heat transport changes into the Atlantic and Indo-Pacific (not shown, see Table for Reviewer below). In a fully
 coupled framework, Liu et al. (2018) showed that in response to quadrupled atmospheric CO₂ concentrations,
 the poleward-strengthened westerlies displace and intensify the Southern Ocean's meridional overturning
 circulation which results in anomalous heat transport divergence at 60°S and increased surface heat fluxes
 while the opposite was shown for 45°S. In our wind-only simulation, we see strong heat transport divergence
 at almost all latitudes of the southern Indian and Pacific sectors, while heat converges in the Atlantic sector
 between 60°S-45°S (Extended Data Figure 5b), likely because the Southern Ocean surface wind trends in
 JRA55-do are strongest in the Indian and Pacific sectors (see Figure for Reviewer below). We agree with Liu
 et al. (2018), that wind stress changes are likely the primary drivers of ocean heat content change in the wind-
 only simulation (through their induced SST changes), rather than the direct wind-speed related turbulent heat
 flux change.”

	Southern Ocean into Atlantic Ocean	Southern Ocean into Indo-Pacific Ocean	Indonesian Throughflow
Advective flux	2.9	1.3	0.9
Submesoscale eddy parameterisation	0.0	0.0	0.0
Gent-McWilliams (1990) eddy parameterisation	0.0	0.2	0.0

Neutral diffusive parameterisation flux	0.1	0.2	0.0
Total	3.0	1.6	0.9

**Table for Reviewer. The components of total ocean heat transport across the transects separating the**
 **ocean basins in Fig. 3b.** The ocean heat transport rates from the resolved advective and parameterised
 submesoscale, Gent-McWilliams (1990) and neutral diffusive fluxes (10^{21} J year⁻¹) are shown across the
 different transects. The values are rounded to one decimal point and as a result not all cases sum up to the
 Total value shown at the bottom row. Some terms are shown as 0.0 although their magnitude is non-zero (just
 $< 0.05 \times 10^{21}$ J).

 **Figure for Reviewer. Wind trends during 1972-2017 in the JRA55-do reanalysis product.** The values
 are shown in units of m s⁻¹ year⁻¹ and the shading shows the zonal component. The vector scale is shown at
 the top right of the panel.

References not cited in the manuscript

1. Couldrey, M.P., Gregory, J.M., Boeira Dias, F., Dobrohotoff, P., Domingues, C.M., Garuba, O., Griffies,
 S.M., Haak, H., Hu, A., Ishii, M. and Jungclaus, J., 2021. What causes the spread of model projections of
 ocean dynamic sea-level change in response to greenhouse gas forcing?. *Climate Dynamics*, 56, 155-187.

2. Liu, W., Fedorov, A.V., Xie, S.P. and Hu, S., 2020. Climate impacts of a weakened Atlantic Meridional
 Overturning Circulation in a warming climate. *Science Advances*, 6, eaaz4876.

3. Liu, W., Lu, J., Xie, S.P. and Fedorov, A., 2018. Southern Ocean heat uptake, redistribution, and storage
 in a warming climate: The role of meridional overturning circulation. *Journal of Climate*, 31, 4727-4743.

4. Winton, M., Griffies, S.M., Samuels, B.L., Sarmiento, J.L. and Frölicher, T.L., 2013. Connecting changing
ocean circulation with changing climate. *Journal of climate*, 26, 2268-2278.

These references have now been included either in our discussion of the North Atlantic subpolar gyre/AMOC
feedback, or in the discussion of changes in the Southern Ocean's overturning circulation.

Ref.: NCOMMS-22-03243

All line and figure numbers in blue point to the manuscript with tracked changes active.

-----
**Reviewer #2:**
-----

This paper presents a 1972-2017 hindcast of ocean heat uptake and details where ocean heat content has
been stored and transported. It goes without saying improved estimates of these metrics are of great scientific
importance and worthy of publication in Nature Communications. This especially is the case for the pre-Argo
era where observations are sparse. However there presently exists a large number of published observations,
re-analysis and modeling-based estimates of these quantities (as the authors reference in the paper).
Previous work also reaches the same key conclusion that the Southern Ocean dominates heat uptake and
storage (e.g., Frolicher et al. 2015). So, one important question is why should we place emphasis on the
results and conclusions presented by the authors here over the previous modeling work? Here the paper
stumbles. It certainly emphasizes what is poor about previous efforts, but in the main text itself presents very
little convincing arguments about what is superior about using re-analysis to drive the model and the decadal
spin up technique. For that matter, in just reading the main text one would have little idea about what the new
technique even entails (see my comment on line 45). Therefore, I would argue that presenting a strong case
for the superiority of their hindcast compared to previous work is the lynchpin of the study, as how much we
care about the excellently presented results that follows depends upon the novelty and superiority of these
techniques used to create them. In the methods sections and the SI a case is built, but I think that this
information has to be presented more as a result in the paper, not just in the methods section.

We thank the reviewer for their useful and informative comments that have helped to improve the manuscript.
Below we address each comment individually. We agree that a novel aspect of this paper is the experimental
design and so we have now made a series of changes, listed in more detail below, in order to better emphasize
these advances in the paper introduction and main text.

In the abstract (line 15ff.), we state:

“Here, we equilibrate a reanalysis-forced ocean-sea ice model using a novel spin-up approach that
improves on previous approaches, and investigate recent OHU trends associated separately with atmospheric
wind trends, thermodynamic properties (temperature, humidity and radiation) or both.”

... and we moved parts of the Methods into the introductory section (lines 57ff.) to highlight key advantages
of our study and why it is important:

“In this study we address these limitations by introducing a new spin-up protocol for global ocean-sea
ice models and illustrate its benefits using the ACCESS-OM2 ocean-sea ice model (Kiss et al., 2020). The
spin-up is performed using repeat decadal cycles of the JRA55-do atmospheric reanalysis forcing over the
period 1962--1971, the decade prior to the recent rapid acceleration in OHU (Schuckman et al. (2020), Fig.1
and Methods). By equilibrating the model this way, there are no longer large initial shocks at the beginning of
each cycle when the forcing transitions from 2018 back to 1958. Furthermore, model drift can be accounted
for by subtracting the linear trend from a parallel control repeat-decade simulation (Fig. 1b, c). This new spin-

up procedure leads to improvements in the OHC trajectory from the 1960s onward (Extended Data Fig. 2a).
Using this new approach we estimate ocean heat uptake, transport and storage trends over the last 50 years
in a simulation where all forcing fields evolve over time (the “full forcing” simulation). We also perform
additional simulations that isolate changes driven by trends in surface winds only, trends in atmospheric
thermodynamic properties only (i.e., atmospheric temperature, humidity and radiation changes, hereafter
referred to as the “thermal” experiment) or regional atmospheric trends only. In these experiments, any
unperturbed forcing fields continue cycling through the repeat 1962-1971 forcing (Methods).”

To further highlight our new spin-up and its advantages, we have also moved the Figure depicting the
experimental design of the spin-up into the main manuscript as the new Fig. 1.

Another strength of this paper lies in the subset of both mechanistic (thermal vs wind) and regional hindcast
runs. In particular, the Southern Ocean only forced experiment basically reproducing the global heat storage
of the global forced experiment strikes me as a very important result. However, the fact that the study contains
the regional experiments barely gets mentioned in the abstract. So perhaps more emphasis on the
experiments performed in both the abstract and introduction will also differentiate this work more from past
modelling studies.

Yes, we agree. We have now included, as best as possible within the abstract word limit, information on the
new spin-up and the different perturbation experiments we run (lines 15ff.):

“Here, we equilibrate a reanalysis-forced ocean-sea ice model using a novel spin-up that improves on
earlier approaches to investigate recent OHU trends basin-by-basin and associated separately with surface
wind trends, thermodynamic properties (temperature, humidity and radiation) or both.”

In the abstract, we also highlight the simulation with only regionally applied atmospheric trends over the
Southern Ocean (line 23ff.):

“[...], while Southern Ocean forcing trends can account for almost all of the global OHU.”

In the introductory section, we present more information about the spin-up, its advantages and the
perturbation simulations(see the paragraph starting line 57 and also our comment above).

Both of these comments are largely stylistic, and I feel that the material submitted (i.e., main text+SI) can
largely be altered to address them without the need for additional experiments to be performed. Aside from
the caveats associated with the fact that the results are produced by an ocean model driven by a reanalysis
product, the scientific analysis and presentation seem accurate and clear. To that end I would recommend
acceptance after minor revisions.

Comments:

Line 41: If it is independent, why do you care about removing drift later on.

To avoid confusion, we removed: “the model’s oceanic state is independent of the initial conditions” as this
is only true once the model is completely free of drift, which is not the case here.

The important improvement in our approach is that in the OMIP-2 protocol, there is no parallel running
control simulation and drift cannot be accounted for, whereas here we have a parallel control run and we
account for model drift by removing the linear trend in this simulation (see Fig. below).

Fig. for Reviewer. Comparison between the five-cycle OMIP-style spin-up and the new repeat decade forcing spin-up with a parallel running control simulation.

Line 45: Evidence is needed in the main text to explain and show that the new spin-up technique of using
repeat decades is better than previous efforts. After all, the quality of the results presented in this paper
depend upon this.

Yes, we agree and have moved parts of the Method section into the introductory section in lines 63ff. to show
how our new spin-up technique improves on the previous approach:

“By equilibrating the model this way, there are no longer large initial shocks at the beginning of each
cycle when the forcing transitions from 2018 back to 1958. Furthermore, model drift can be accounted for by
subtracting the linear trend from a parallel control repeat-decade simulation (Fig. 1b, c).”

Line 56: Reaches 2.40E23J by which year? The term ‘observational record’ is rather vague. Which
observations exactly?

Thank you. This sentence has now been reworded to read:

“The observations of upper 2000 m global OHC in Levitus et al. (2012) reach 2.40×10^{23} J in 2017
relative to the 1972–1981 baseline (dashed red line, Fig. 1a).”

Line 62: Please define in the main text what is meant by full forcing. At this stage of reading, I have no idea
what reanalysis dataset is being used or what is even meant by forcing (perhaps though this is more clearly
stated at line 119). I have no problem about things being in the method section, but the novelty of the paper
and results themselves lie in these techniques. A more solid description is needed in the main text (to repeat
my comment above!)

Thank you for the comment. We have now included this information in the introductory section of the
manuscript (paragraph starting line 57). There, we specify that we use the atmospheric reanalysis data set

JRA55-do to force our model and we have added information about the specific perturbation simulations from
the Methods section in this paragraph:

“Using this new approach we estimate ocean heat uptake, transport and storage trends over the last
50 years in a simulation where all forcing fields evolve over time (the “full forcing” simulation). We also perform
additional simulations that isolate changes driven by trends in surface winds only, trends in atmospheric
thermodynamic properties only (i.e., atmospheric temperature, humidity and radiation changes, hereafter
referred to as the “thermal” experiment) or regional atmospheric trends only. In these experiments, any
unperturbed forcing fields continue cycling through the repeat 1962-1971 forcing (Methods).”

Line 66: Could not the improvement be coincidental. Why does it improve?

The main evidence can be seen in the more realistic ocean heat content anomalies in the 1960s, while the
OMIP-2 models show strong cooling in the first two decades due to the initial shock of returning from the warm
2018 state instantaneously back to 1958 (Extended Data Fig. 2). In the OMIP-2 models, it is unclear how long
into the record this cooling has an impact, but it appears to be at least 2 decades, and due to slow interior
ocean subduction and mixing processes, probably somewhat longer.

The other advantage of our parallel running control simulation is that it allows for a clear accounting of model
drift, something that is not possible in OMIP-2. In OMIP-2, it is not clear if drift should be removed by fitting a
linear trend over the last two cycles, three cycles or if it should be accounted for in a different way (see the
Figure from above).

Line 77: How do the CMIP5 models go to 2017 or for that matter CMIP6 which stops in 2014?

The historical CMIP6 simulations end in the year 2014. We have now extended the CMIP6 ocean heat content
time series in Fig. 1 to the year 2017 using model projections from the SSP5-8.5 scenario for the years 2014-
2017. We chose the SSP5-8.5 projection as the difference to other projections (e.g., SSP4-6.0) over these
three years is minimal.

Figure 1: Levitus 2020 (line 75) or 2012 (figure key)?

Thank you for pointing out this mistake. The correct reference is Levitus et al. (2012).

Figure 2: Just to confirm I see just one blue line here demarking the Southern Ocean (not lines as the caption
says). Should the continental land masses be used to separate the other ocean basins?

We have changed the thick blue lines to black lines and adjusted the caption for Fig. 2 in the following way:

“[...] show the total area integrated trends over a particular ocean basin with the boundaries set by the
black lines across the Southern Ocean, the Indonesian Throughflow, the Bering Strait and the continental
land masses.”

Line 125: I agree with the sentiment here, but you are still subject to the quality of the reanalysis surface fluxes
and (for example) if they properly capture radiative forcing trends due to greenhouse gases and aerosol. Have
you thought about what might happen if you used ERA5 or another product?

Thank you for this comment. We agree that this is a potential draw-back of our study and that our results
 may be dependent on the atmospheric reanalysis product used. Most ocean-sea ice models including
 ACCESS-OM2 are currently configured to be forced by JRA55-do (e.g. the OMIP-2 protocol). There is
 ongoing work in our modelling group to develop a configuration forced by the ERA5 product. However,
 these simulations are not yet available. Yes, we would expect some differences when forcing with either
 ERA5 or another reanalysis product as there are substantial global and regional variations in the net surface
 heat fluxes (as shown in Fig. 2 of Valdivieso et al. (2017)).

**Fig. 2 in Valdivieso et al. (2017).** Global mean heat fluxes averaged over the 17-year period (1993–2009)
 along with their interannual standard deviations over this period.

We have added more information in the section that highlights limitations in our study in lines 256ff:

“Uncertainties also arise from inherent uncertainties in the reanalysis product used, including the
 reliability of the implied radiative heat flux trends due to both greenhouse gases and aerosols, which remain
 poorly constrained in observations.”

Reference:

- - Valdivieso, M., Haines, K., Balsaseda, M. *et al.* An assessment of air–sea heat fluxes from ocean and
 coupled reanalyses. *Clim Dyn* **49**, 983–1008 (2017). <https://doi.org/10.1007/s00382-015-2843-3>

Line 178: I think the conclusion here supports my claim that you should devote at least some of the main text
 (not methods) on explaining and evaluating the spin-up technique.

Yes, we agree. As discussed in responding to the reviewer’s main point above, we have now moved parts of
 the method section into the introductory section of the manuscript.

Line 243: This is an example of some text, which details the improvement, that would be better off in the main
text.

Yes we agree. We moved the majority of this paragraph to the last part of the introduction section at lines
57ff.:

“By equilibrating the model this way, there are no longer large initial shocks at the beginning of each
cycle when the forcing transitions from 2018 back to 1958. Furthermore, model drift can be accounted for by
subtracting the linear trend from a parallel control repeat-decade simulation (Fig. 1b, c). This new spin-up
procedure leads to improvements in the OHC trajectory from the 1960s onward (Extended Data Fig. 2a).”

REVIEWERS' COMMENTS

Reviewer #1 (Remarks to the Author):

The authors have satisfyingly addressed all my comments. I therefore recommend publication.

Reviewer #2 (Remarks to the Author):

I thank the authors for addressing my comments in detail. I am satisfied with their responses and happy for the paper to be published in its present form.

The authors have satisfyingly addressed all my comments. I therefore recommend publication.

Great!

I thank the authors for addressing my comments in detail. I am satisfied with their responses and happy for the paper to be published in its present form.

Awesome!